# CATEGORICAL MODEL OF NEURAL NETWORKS

**Georgy Tolokonnikov**
FNAC VIM RAS
Moscow, Russia
admcit@mail.ru

## ABSTRACT

Neural networks are modeled as categorical systems, the theory of which is developed by the author, which is useful not only from the conceptual point of view of identifying the systemic nature of neural networks, but also for specific issues. In particular, the work provides a categorical justification for the well-known formulas of S. Osovsky used in the method of backpropagation of error. The theory of categorical systems allows one to naturally model (which is done in the work) not only traditional artificial neural networks of arbitrary topology, but also networks of living neurons, which in addition to spike communication have several dozen other types of cellular communication, and also allows one to model network structures similar to higher categories. Polycategories were introduced in 1975 by Szabo as a set of polyarrows, the composition of which is defined similarly to the composition of arrows and multiarrows in categories and multicategories. For modeling neural networks such a connection of polyarrows is not enough, to some extent this is removed in PROP, introduced by MacLane (and in their varieties in the form of dioperads and others). We replace compositions of polyarrows (including compositions in higher categories) with a new more general type of connections called convolutions, introduce and use categorical splices, from which polycategories are built with an explicit assignment of the history of obtaining polyarrows (analogous to a nerve in categories). Models of categorical gluings are categories, algebraic systems, double categories, PROP and other higher categories, which are considered in the work. A number of new definitions and results are given and some review of works on categorical topics in the theory of systems and neural networks is given.

## 1 INTRODUCTION

The categorical model of neural networks arises naturally in attempts to identify the systemic aspect in them or to consider neural networks from a systemic point of view. It can be considered a generally accepted view that systems theory as a science of sciences (see, for example, [1]), which seems natural if physics is considered a theory of physical systems, biology is considered a theory of biological systems, sociology is considered a theory of social systems, and so on. Neural networks are also modeled as corresponding systems, which is useful not only from the conceptual point of view of identifying the systemic nature of neural networks. Such modeling turns out to be useful for specific issues, in particular, the substantiation of the well-known formulas of S. Osovsky, used in the backpropagation method [2], considered in the work. The attempts at substantiation given in [2] at the engineering level of rigor can now be replaced by a strictly mathematical substantiation of the said formulas within the framework of the categorical model of neural networks in the form of convolutional polycategories. For convolutional polycategories there are three types of duality (unlike ordinary categories), two of which, when applied to a neural network, yield an object that S. Osovsky called a "conjugate graph". Moreover, the categorical theory of systems that we are developing [4] allows us to naturally model not only traditional artificial neural networks of arbitrary topology, but also networks of living neurons, which, in addition to spiking communication, have several dozen other types of cellular communication (the accounting of non-spiking communication of neurons in artificial neural networks began, for example, in [3]), and also allows us to model network structures similar to higher categories [5]. The theory of functional systems by P.K. Anokhin [6] has a categorical nature [4], which is reflected even in the name (in contrast, for example, to the systems according to Mesarovich, which are relations on the Cartesian product of a set of inputs and a set of outputs [7]). In the postulates of P.K. Anokhin [6], the construction of a system goes from

the whole (the system-forming factor) to the parts, therefore, it is the categorical language that is adequate to the theory of systems, and not the traditional set-theoretic language, which does not cover, for example, the fact that the set of functional systems is not a set, but a topos, and a non-classical one at that [4]. Another important postulate consists in the requirement to construct a system and its theory, proceeding exclusively from the system-forming factor, which is also one of several basic postulates of the categorical theory of systems, which serves as a formalization and development of the general theory of functional systems by P.K. Anokhin. The systemic view of science that we propose is as follows. Science is a system with a system-forming factor in the form of the task of "building a theory of the subject of science", that is, a set of true (in some sense) statements about the subject of science. Initially, there is nothing except the subject of science and the researchers themselves who have undertaken to build the said theory. Relying only on the system-forming factor, researchers (how they do it is of no interest to us) build an alphabet, words in it, a language with statements, the concept of the truth of statements, tools for determining the truth of statements, and logic for proving the truth of those statements (for example, statements of community) for which the tools are insufficient. In other words, the object of study dictates everything, including the logic of the theory, while excluding, for example, the use of classical logic that is in no way justified based on the object. Thus, there are as many logics as there are objects... A brilliant example of a systematic, in our sense, construction of a theory is the actual construction of a theory of words in alphabets in the book by A.A. Markov [8] (words in alphabets and constructive operations used to write out words are an object, constructive logic without the law of the excluded middle, constructed in [8] at an informal level, including, in particular, several negations and implications, also arose on the basis of reliance on the specified object of study). Our systemic approach is consistent with an important methodological principle of science, formulated in [9]: not to make such a "serious mistake... as adjusting the formulation of a problem to conventional methods of solution, rather than searching for methods corresponding to the original substantive problem." In the terminology of P.K. Anokhin's functional systems, "formulation of the original substantive problem" is a conscious formulation of a system-forming factor. "The search for methods corresponding to the original substantive problem" as a system-forming factor includes, in our interpretation, also a search for logic appropriate to the subject of study (rather than using ad hoc, for example, classical logic) to build a theory of this subject of study. We use the systemic approach in our sense further when discussing constructivity in algebra. In the categorical theory of systems, the generally accepted set-theoretic paradigm, in which objects of the objective world, virtual reality, etc., are modeled by sets and subsets, is replaced by a systems paradigm, according to which the modeling of the said objects is carried out by systems and their collections. Thus, a neural network is initially considered as a system (in our case, a categorical system), for which a number of categorical system properties are fulfilled, which are considered in this paper. Polycategories were introduced in 1975 by Szabo [10] as a set of polyarrows, the composition of which is defined similarly to the composition of arrows and multiarrows in categories and multicategories. Polyarrows have inputs and outputs, and it is natural to model neurons with them. However, the connections of neurons observed in the brain [11] are significantly richer than the possibilities that the composition in Szabo's polycategories can provide. Szabo's theory of polycategories finds applications [12], but is very complex, and working with them "manually", as Szabo does according to R. Garner, encounters problems [13]. For the case of symmetric polycategories, R. Garner constructed their representation in the form of monads in a suitable two-sided Kleisli bicategory. The representation constructed by G. Garner generalizes a similar well-known representation of multicategories in the form of monoids in the special categories [14-16]. An attempt at such a generalization for arbitrary polycategories in [17] did not lead to the final construction of the indicated representation of polycategories. Szabo introduced polycategories for problems of logic, where the inference rules are polyarrows in Gentzen's approach (conjunctions of premises are translated into disjunctions of formulas) and their connections can be reduced [13] to connections between themselves using one output of the first and one input of the other polyarrow. This limitation in the methods of connecting polyarrows is to some extent removed in PROPs introduced by MacLane [18] (and in their varieties, dioperads, etc., see [19], as well as the review by Markl M. Operads and PROPs, 2006, arXiv:math/0601129v3), which are used in the categorical approach to networks in [20]. In our approach, we proceed from the needs of modeling the connections of neurons in the brain, for which there are not enough connections in the form of the studied compositions of polyarrows in the Szabo and PROP polycategories, and systems theory, when it is necessary not only to assemble a system from future subsystems, but also to decompose it into subsystems. To this end, we replace polyarrow compositions (including compositions in higher categories) with a new, more general type of connections called convolutions, and introduce and

use categorical splices, from which polycategories are constructed with an explicit specification of the history of obtaining polyarrows (analogous to a nerve in categories) [4, 21–25]. The concept of a system in many systems approaches is associated with the concept of a whole; the transition from a whole to parts, as we have already noted, requires the language of category theory during formalization. This idea is discussed in [26] and developed in [27]. However, the cited approaches use traditional category theory; as noted above, categorical splices and convolutional polycategories provide more opportunities for modeling systems. The paper presents a number of new definitions and results, provides proofs of previously announced theorems, and also provides a brief overview of works on categorical topics in systems theory and neural networks.

## 2 CATEGORICAL SPLICES

Let us consider a language that, with additional requirements, can be part of the language of predicate calculus or first-order theory with classical logic. We emphasize that this language can be used in a similar way, for example, for constructive mathematics according to A.A. Markov and in the general system constructive case considered below.

Let us construct an alphabet with variables and, accordingly, functional symbols of projections $x_k{}^{(i)}$, $C^{(j)}$, $i, j, k$ - here and below natural numbers, the choice of $i$ or $j$ means the choice of the type of variable. In the constructive case, natural numbers can be realized, as usual, by a set of dashes. In the usual way, we introduce terms $t^{(i)}$, $v^{(j)}$, $u^{(k)}$, ... constructed from the indicated variables and projections.

We introduce functional symbols $\xi_{ij}$ of the type $(i{\rightarrow}j)$ with properties $\xi_{ij}\ \xi_{ji}\ t^{(i)}{=}t^{(i)}$. . We introduce predicate letters of equality for each sort $=_i$ , as well as equalities between sorts, as designations $t^{(i)}=_{ij}v^{(j)} =^{def}(t^{(i)}=_i \xi_{ij}\ v^{(j)})$. Predicate letters, when interpreted in classical logic, turn into ordinary predicates, and when considering constructive logic, into constructive predicates that correspond to only one truth value "true". For each $i$ we denote $C_i{}^{(j)} = \xi_{ij}\ C^{(j)}\ \xi_{ji}$ and introduce the axioms $C_i{}^{(k)}\ C_i{}^{(j)} =_i C_i{}^{(l)}$. Thus, the functional symbols form an algebra with a multiplication table corresponding to the last axioms. We call basic formulas expressions of the form $t^{(i)}=_{ij}v^{(j)}$ .

We introduce the conjunction sign into the alphabet as a binary functional sign. We call formulas expressions obtained by applying conjunction to basic formulas and any expressions obtained by such application. In order to cover the constructive case, we do not use the concept of the set of all formulas and other concepts where it is necessary to refer to the actual infinity. We postulate the property of commutativity and associativity in conjunction, which is achieved by the absence of brackets in formulas with conjunctions in the usual notation. We also introduce variables $a_k^{(i)}$ with the property $a_k^{(i)} =_i C^{(i)}x_k^{(i)}$ .

For each sort, we introduce a set of functional symbols $\mu_l^{(i)}, \nu_l^{(l)}, ..., l = 1, 2, ...$ , as symbols of $n$-ary operations, $n$=0,1,2,3,... and a set of identities $t_m^{(i)} =_i t_k^{(i)}$ , as terms of the same sort constructed with these symbols in mind.

A *signature* is a set of functional letters of projections and operations $C^{(i)}, \mu_l^{(i)}$.

A category is a set of arrows (if objects are identified, as is often done, with single arrows), which have a visual image of directed segments. Similarly, categorical splice has as its elements visually depicted rows of vertical lines, which we will call combs, as well as the formulas corresponding to them. Thus, the comb of categorical splice defined below consists of an external comb, an external convolution, an internal comb, and an internal convolution. For a given set $x_k^{(i,\alpha)}, C^{(i,\alpha)}, \mu_l^{(i,\alpha)}, ..., =_{i,\alpha,i',\alpha'}, \xi_{ij}^{(\alpha)}$ that is used for the *outer combs*, we enter three copies of it. The set $x'{}_k^{(i,\alpha)}, C'^{(i,\alpha)}, \mu'{}_l^{(i,\alpha)}, ..., ='_{i,\alpha,i',\alpha'}, \xi'{}_{ij}^{(\alpha)}$ is used for *outer convolutions*. The set $\bar{x}'{}_k^{(i,\alpha)}, \bar{C}'^{(i,\alpha)}, \bar{\mu}'{}_l^{(i,\alpha)}, ..., =\!\bar{}'_{i,\alpha,i',\alpha'}, \bar{\xi}'{}_{ij}^{(\alpha)}$ is used for *inner combs* . The set $\bar{x}_k^{(i,\alpha)}, \bar{C}^{(i,\alpha)}, \bar{\mu}_l^{(i,\alpha)}, ..., =_{i,\alpha,i',\alpha'}, \bar{\xi}_{ij}^{(\alpha)}$ is used for the *inner convolutions*.

Definition. A complete base comb is a formula composed of conjunctions of formulas

$$a_k^{(i.\alpha)} =_{i,\alpha} C^{(i,\alpha)}(x_k^{(i.\alpha)}), a'{}_k^{(i.\alpha)} ='{}_{i,\alpha}C'^{(i,\alpha)}(x'{}_k^{(i.\alpha)}),$$

$$\bar{a}_k^{(i.\alpha)} \stackrel{=}{=}_{i,\alpha} \bar{C}^{(i,\alpha)}(\bar{x}_k^{(i.\alpha)}), \bar{a'}_k^{(i.\alpha)} \stackrel{=}{=}'_{i,\alpha} \bar{C}'^{(i,\alpha)}(\bar{x'}_k^{(i.\alpha)})$$

and equalities $\eta_k^{(i,\alpha)} = \theta_m^{(j,\beta)}, \eta_k^{(i,\alpha)}, \theta_m^{(j,\beta)}$ can be any of the symbols present in the previous formulas, the equal sign has the corresponding indices omitted here.

Definition. We distribute the conjunctions for each formula of complete combs across four cells of the 2x2 table. In cell (1,1) we place the subformulas of the outer combs. In cell (1,2) we place the subformulas of the outer convolutions containing letters with strokes. In cell (2,1) we place the subformulas of the inner combs containing letters with overlining and in cell (2,2) we place the subformulas of the inner convolutions containing letters with overlining and strokes:

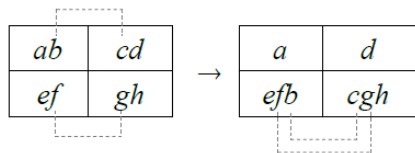

When placing a formula in a table, strokes and overlinings can be omitted; they can be restored in an obvious way if necessary According to the construction of full comb formulas, the equality subformulas contain only those letters of the variables that were encountered in subformulas of the form $a = C(x)$. This allows the equal signs (formulas with equalities of variables) to be excluded from the table, replacing the equality formulas with dotted lines connecting the letters of the variables included in the equalities. Let us define constructive operations $M_t, M_r, M_l, M_b$ specific to categorical splices that transform some full comb formulas into others by moving formulas between the cells of the full comb table.

$M_b$ moves those conjunctions from formulas that have connections (one of which is on the left, the other on the right) to the lower cells from the upper ones

| $ab$ | $cd$ |
|------|------|
| $ef$ | $gh$ |

$\rightarrow$

| $a$ | $d$ |
|-----|-----|
| $efb$ | $cgh$ |

when moving, a change in sorts is set that corresponds to the definition $M_b$.

$M_t$ moves those formulas that have connections (one of which is on the left, the other on the right) to the upper cells from the lower ones

| $ab$ | $cd$ |
|------|------|
| $ef$ | $gh$ |

$\stackrel{M_t}{\rightarrow}$

| $abf$ | $gcd$ |
|-------|-------|
| $e$ | $h$ |

when moving, a change in sorts is set that corresponds to the definition $M_t$.

$M_l$ moves vertically and internally connected pairs from right cells to left cells (movement direction to the left)

| $ab$ | $cdm$ |
|------|-------|
| $ef$ | $ghn$ |

$\stackrel{M_l}{\rightarrow}$

| $abdm$ | $c$ |
|--------|-----|
| $efh$ | $gn$ |

when moving, a change in sorts is set that corresponds to the definition $M_l$.

$M_r$ moves connected pairs from left cells to right cells (movement to the right)

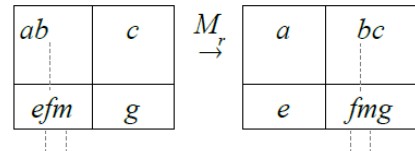

when moving, a change in sorts is set that corresponds to the definition $M_r$.

The constructive operation Ev is introduced, and its inverse $Ev^{-1}$ based on the functional symbols of the operations $\mu_l^{(i,\alpha)}$ and the placement of equal signs in the appropriate places. The application of the convolution present in the cell (1,2) of the comb table consists of successive application of the operations $M_b$, $M_l$, Ev. The inverse operation of the indicated one consists of successive application of the operations $Ev^{-1}$, $M_r$, $M_t$ .

We introduce the operations $P_{k,l}, k, l = 1, 2$, which project the full comb into its parts, translates the full base comb into the contents of the $(k, l)$-cell of the table.

Definition. A complete comb is obtained from a complete base comb in which the variables x (with indices) are replaced by terms of the formal language of the signature from the functional letters, operations and projections. A categorical splice is a set of all possible combs.

Note. In the constructive case, it is not possible to talk about a set of all possible combs of one kind or another; here, they usually mean a property written out as a suitable formula. The constructive set is understood to mean the specified formula, and belonging to the set is thought of as a synonym for the presence of a given property in a constructive object.

Definition. The procedure of applying the convolution, given above for complete base combs, is literally applicable to complete combs and gives an inductive construction of ***formula terms*** :

a complete base comb and its $P_{k,l}$-projections are formula terms;

if there is a formula term, then applying the convolution to it is also a formula term, as are its $P_{k,l}$-projections, and there are no other formula terms.

A formula term, like a complete comb, is decomposed into an outer term, an outer convolution term, an inner convolution term, and an inner term. A formula $a_k^{(i)} = C^{(i)} x_k^{(i)}$ is called an ***elementary external comb***, respectively, formulas $a'^{(i)}_k = C'^{(i)} x'^{(i)}_k$, $\bar{a}'^{(i)}_k = \bar{C}'^{(i)} \bar{x}'^{(i)}_k$, $\bar{a}^{(i)}_k = \bar{C}^{(i)} \bar{x}^{(i)}_k$ are called an ***elementary external convolution, an elementary internal comb, an elementary internal convolution***.

Let a complete comb be given. The outer and inner combs, outer and inner convolutions ( $P_{k,l}$-projections of the complete comb) are called connected comb and convolutions if the elementary combs and convolutions that make them up are connected by dotted lines. The dotted lines of the complete comb that connect different cells of the table are not taken into account in this case.

Following Hatcher, we introduce graphic notations similar to those used in his book [28], for example, for the formula $C^{(1)}(x_i^{(1)}) = a_i^{(1)} \wedge C^{(2)}(x_j^{(2)}) = a_j^{(2)} \wedge x_i^{(1)} = x_j^{(2)}$ we have

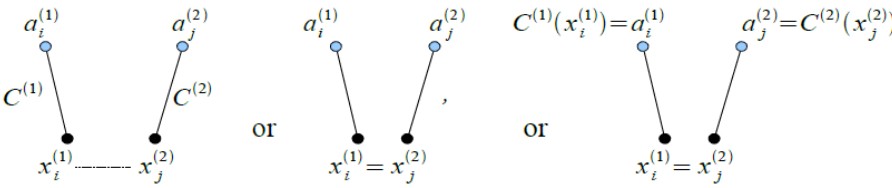

In the bottom row of the comb table, we will mirror the graphic images of formulas (together with the dotted lines of equalities) from bottom to top, for example:

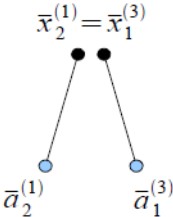

For clarity, we will depict different names or areas connected by equality closer to each other on the graph (for the figures provided, this agreement is taken into account due to $x_i^{(1)} = x_j^{(2)}, x_2^{(1)} = x_1^{(3)}$ ).

The considered constructive operations on formulas define a certain formal predicate, which with a suitable modification can be reduced to a three-place or multi-place predicate $K(x, y, z, t, ...)$. In it, the variables run through connected combs and convolutions. We will consider specific examples for convolution polycategories and multicategories below.

Let us construct a categorical splice for the traditional category theory; the use of convolutions will give in the gluing a "history" of its implementation, which is important for systems theory, since it models the formation of a system from subsystems. For the case of small categories C, the specified "history" for different combs turns out to be the nerve of category C.

To represent the composition of category arrows by combs of categorical splice, we will need an eight-base theory. We introduce variables $x_i^{(1)}, x_i^{(2)}, x_i^{(3)}, ..., x_i^{(8)}$ , functional letters $C^{(1)}, C^{(2)}, C^{(3)}, ..., C^{(8)}$, , functional symbols $\xi_{i,j}, i \neq j, i, j = 1, 2, ..., 8$ of bijections, respectively, of the form $(j \to i)$ , equality predicates $=_1, =_2, ..., =_8$ in each sort and the equality predicate = for different sorts. We will assign varieties to the cells of the table as follows

| 1,2 | 3,4 |
|-----|-----|
| 5,6 | 7,8 |

When moving sorts from right to left (and vice versa), the replacement of sorts has the form $3 \Leftrightarrow 1, 4 \Leftrightarrow 2, 7 \Leftrightarrow 5, 8 \Leftrightarrow 6$ , when moving sorts from top to bottom (and vice versa), we have the following replacement of sorts $1 \Leftrightarrow 5, 2 \Leftrightarrow 6, 3 \Leftrightarrow 7, 4 \Leftrightarrow 8$ . In the letter designations we will retain the strokes and overlines, which are convenient for determining in which cell of the table the letter is located.

| $x_i^{(1)}, x_i^{(2)}, C^{(1)}, C^{(2)}$ | $x\,'^{(3)}_i, x\,'^{(4)}_i, C\,'^{(3)}, C\,'^{(4)}$ |
|---|---|
| $\bar{x}_i^{(5)}, \bar{x}_i^{(6)}, \bar{C}^{(5)}, \bar{C}^{(6)}$ | $\bar{x}\,'^{(7)}_i, \bar{x}\,'^{(8)}_i, \bar{C}\,'^{(7)}, \bar{C}\,'^{(8)}$ |

We introduce variables $a_k^{(i)}, k = 1, 2, ...$ with the property $a_k^{(i)} =_i C^{(i)}x_k^{(i)}, k = 1, 2, ....$ The predicate corresponding to the partial operation of composition $\mu x_i^{(1)}x_j^{(1)} = x_i^{(1)} \circ x_j^{(1)}$ (first-sort variables are multiplied) is denoted by $P = P^{(1,1,1)}(x_i^{(1)}, x_j^{(1)}, x_k^{(1)})$ , using this predicate and bijections one can define a predicate $P^{(a,b,c)}(x_i^{(a)}, x_j^{(b)}, x_k^{(c)}, a, b, c = 1, 2, 3, ..., 8)$ for any combination of sorts. For example, $P^{(1,2,4)}(x_i^{(1)}, x_j^{(2)}, x_k^{(4)}) = P^{(1,1,1)}(x_i^{(1)}, \xi_{1,2}x_j^{(2)}, \xi_{1,4}x_k^{(4)})$ , in particular, $P^{(2,2,2)}(x_i^{(2)}, x_j^{(2)}, x_k^{(2)}) = P^{(1,1,1)}(\xi_{2,1}x_i^{(1)}, \xi_{2,1}x_j^{(1)}, \xi_{2,1}x_k^{(1)})$ .

External base combs consist of formulas of the form $C^{(1)}(x_k^{(1)}) = a_k^{(1)} \wedge C^{(2)}(x_m^{(2)}) = a_m^{(2)} \wedge x_k^{(1)} = x_m^{(2)}$. Their graphic notation (the dotted line indicates the presence of an equal sign in the atomic formula $x_k^{(1)} = x_m^{(2)}$, sometimes it is convenient to replace the dotted line with an equal sign or to explicitly indicate formulas from the conjunctions) has the form

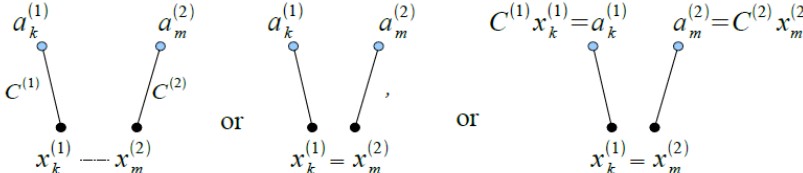

The necessary external compositions have the form $a'^{(3)}_i = C'^{(3)}(x'^{(3)}_i) \wedge a^{(4)}_j = C'^{(4)}(x'^{4}_j) \wedge x'^{(3)}_i = x'^{(4)}_j \wedge a'^{(3)}_i = a'^{(4)}_j, i, j = 1, 2, ...$

or in graphic notations

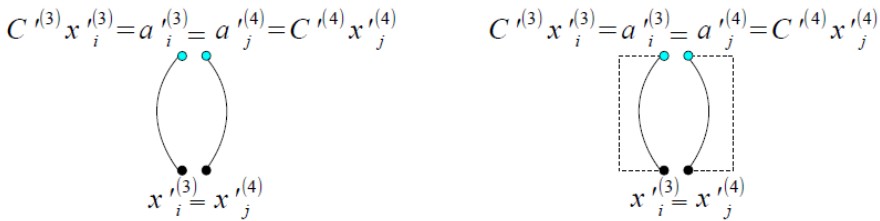

When interpreted in Set or another category, $x_i, x_j$ will go into the arrow names, for compositions in the case of categories, arrows id with $x_i = Cx_i = a_i, x_j = Cx_j = a_j$ should be taken, this option is shown in the diagram on the right. Let us proceed to the formulation of the procedure for carrying out the convolution corresponding to the composition of the category arrows.

Let there be two connected combs (they correspond to two arrows), their product is defined by the convolution procedure, we define equalities according to the table and graphic notation given below (re-designations are introduced to shorten $a_1 = a_1^{(1)}, b_2 = a_2^{(2)}, b_3 = a_3^{(1)}, c_4 = a_4^{(2)}, b_5 = a_5^{(3)}, b_6 = a_6^{(4)}$. When postulating the convolution and the equalities involved in the procedure of its application (dashed lines in the graphic notations), subformulas $a_2^{(2)} = a'^{(3)}_5, a_3^{(1)} = a'^{(4)}_6$ appear that lead to the requirement (reflects the cases of the possibility of applying the composition) $a_2^{(2)} = a_3^{(1)}$. The equality $C^{(1)}x_1^{(1)} = C'^{(2)}x'^{(2)}_2$ gives $C_j^{(a)}C_k^{(a)}x'^{(a)}_i = C_k^{(a)}x'^{(a)}_i, a, j, k = 1, 2$, that is, we obtain an algebra for (the upper indices of C are the same)

| · | s | t |
|---|---|---|
| s | s | t |
| t | s | t |

As a result, we have everything necessary to carry out the convolution procedure, we write out the following complete comb of the considered categorical splice

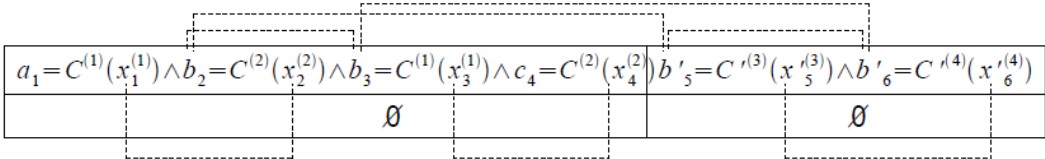

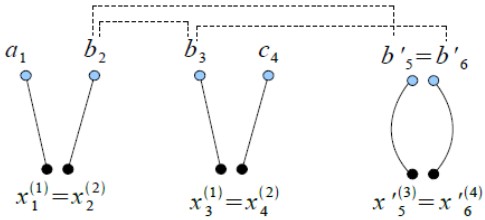

We will immediately consider the convolutions for k combs (representing individual arrows for the category). We will introduce the notation $a_i x_i$ for $a_i^{(j)} =_j C^{(j)} x_i^{(j)}$, j=1,5 and $x_i a_i$ for $a_i^{(j)} =_j C^{(j)} x_i^{(j)}$ for j=2,6.

The k-1 available external convolutions of the form $a_i x_i^{(3)} \wedge x_{i+1}^{(3)}$ can be applied to the comb (implied $x_{2m-1} = x_{2m}$ )

| $a_1 x_1 \wedge x_2 a_2 \wedge a_3 x_3 \wedge x_4 a_4 \wedge a_5 x_5 \wedge x_6 a_6 \wedge ... \wedge a_{2k-1} x_{2k-1} \wedge x_{2k} a_{2k}$ | Ø |
|---|---|
| Ø | Ø |

which we will do.

We apply the first convolution.

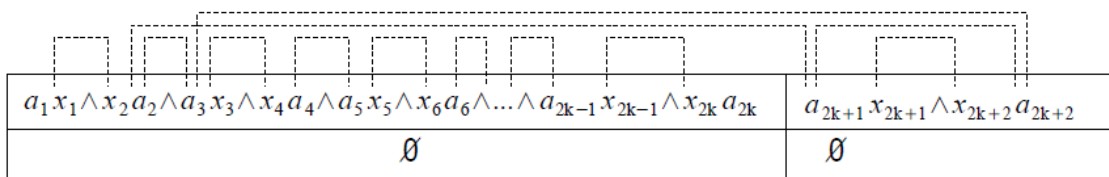

We apply the operation $M_b$, we get

We apply the operation , we get

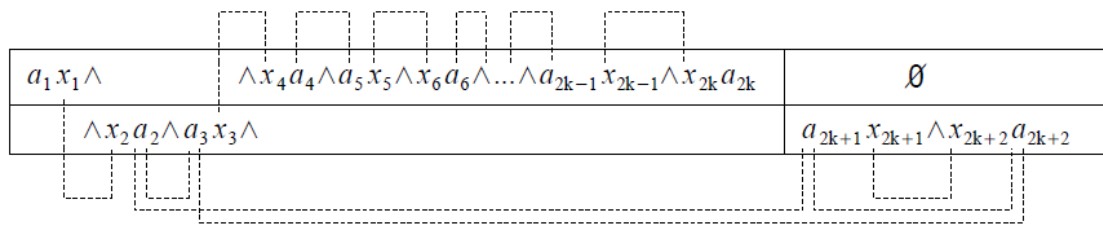

Now we apply the operation $M_l$, we get

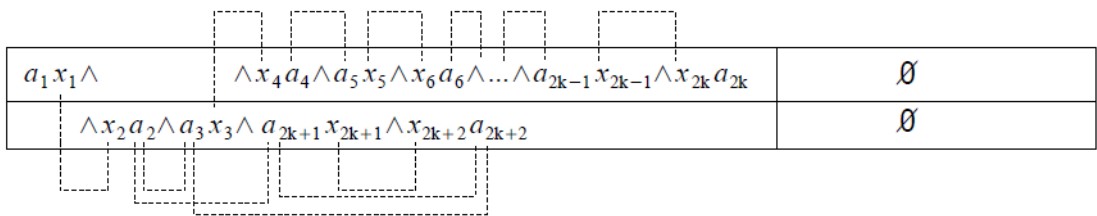

It remains to apply the operation Ev. With the help of the predicate corresponding to the composition , the constructive operation Ev transforms the previous formula into the following .

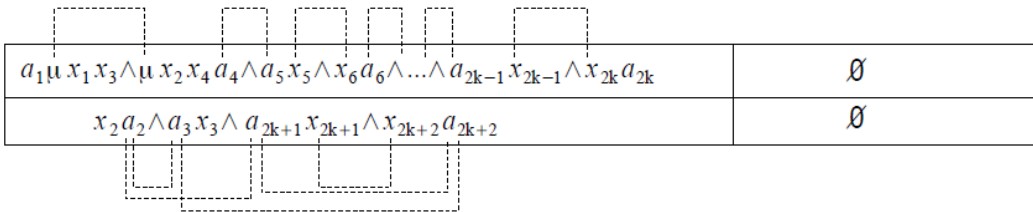

Note. When interpreting a formula, in addition to choosing a composition predicate, it is necessary to determine the choice of a function name for $\bar{x}_{2k+1}^{(5)} = \bar{x}_{2k+2}^{(6)}$ , for the case of categories, this choice is unambiguous in the form of an identity function (unit arrow) id. We will make this choice at the last step of applying folds.

At the second step, we supplement the resulting comb with another convolution $a_{2k+3}x_{2k+3} \wedge x_{2k+4}a_{2k+4}$ and apply the convolution.

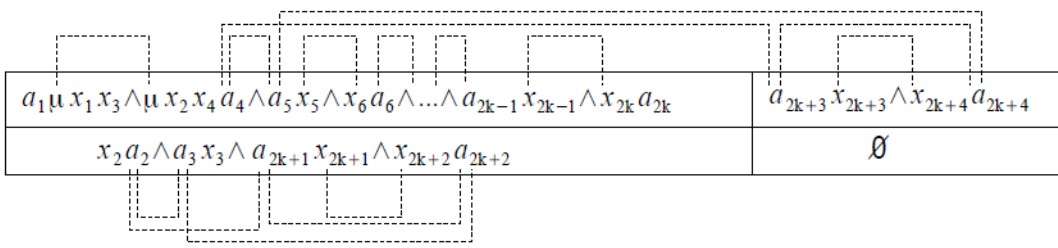

Applying the operations $M_b$ , $M_l$ , and Ev similarly, we obtain

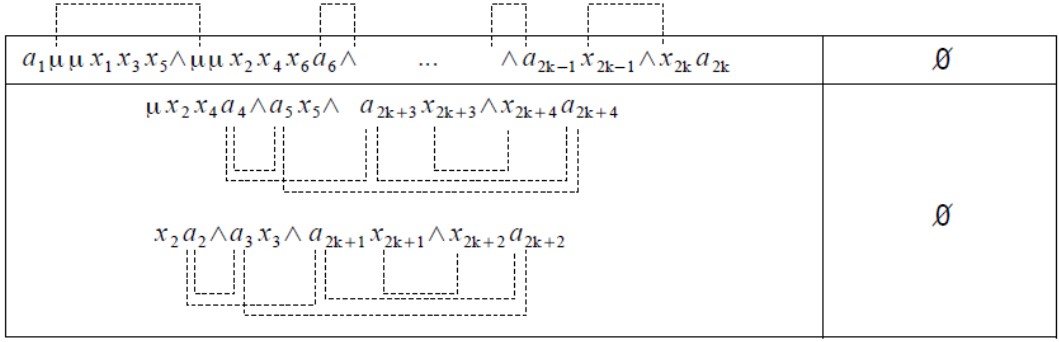

After performing k-1 convolutions we obtain

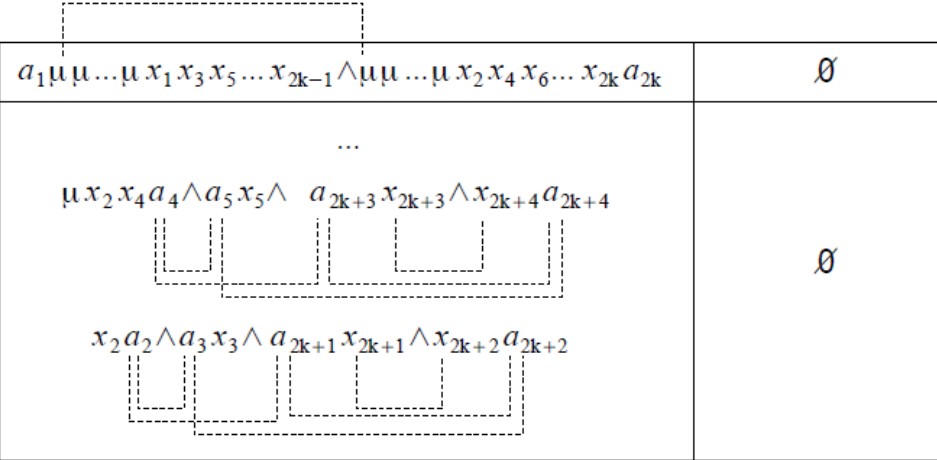

The above derivation contains, in particular, the corresponding derivation for the case of ordinary categories, which can be seen by narrowing it down as follows. We include the language under consideration in the language of predicate calculus with the axioms of category theory given by Hatcher [28].

The choice of the function name for for m=1, 3, ... , as indicated in the note, for the case of categories is reduced to the name id - the identity function (unit arrow). Thus, for the case of categories we obtain a comb obtained from the previous one by replacing with id and redesignating into the usual notation of the composition operation. The calculation carried out without changes is also carried out for this case. Taking the P1,1 -projection of the full comb as the final result, we obtain exactly the composition of k arrows of the category. The interior of the obtained comb, obviously, corresponds one-to-one to the appropriate element of the nerve of the category. If a model of formal category theory is given, for example, a specific category, then the above calculations are also carried out in this case without changes. Summarizing the calculations and reasoning, we obtain the following theorem.

Theorem. Let an arbitrary category be given as a model of a first-order formal category theory, then there exists an explicitly defined categorical splice, the outer part of each comb of which defines the composition of a certain set of arrows of the category, and the inner part of the specified comb defines the corresponding element of the nerve of the category.

In the axiomatics of category theory given by Hatcher [28], the operation of composition is introduced by the predicate K(x,y,z) ("z is a composition of x and y"). A similar formal predicate can be introduced for the operations performed K(s,x,y,z) = "z is the result of applying the convolution s to connected combs x and y". Due to the limited space of the article, we omit the clarification of this procedure.

For the theory of categorical systems, it is the categorical splice that gives in the form of a suitable comb not only the type of the system, but also the "history" of its formation from subsystems, contained in the inner part of the specified comb of the splice, while there is an unambiguous procedure for dividing the system into subsystems. In algebra (theory of algebraic systems [27], model theory), models are understood as interpretations of formal theories of the first and other orders in sets or objects of other categories besides the category of sets. Such models may use logic different from classical logic, such as models in toposes with intuitionistic logic. In our case, we have interpretations of the theory of splices in formal theories with a further transition to the indicated models.

We will call the indicated interpretations of the theory of splices in formal theories formal models. In this terminology, the first part of the given theorem states that there exists a categorical splice for which the first-order category theory according to Hatcher [28] is a formal model.

## 3 Constructivity in Algebra

Systematic study of constructive algebras was started by A.I. Mal'tsev [29] with the introduction of this concept itself in the form of numbered algebras: "The concept of constructivity needs to be clarified, and this clarification can be done in a variety of ways, starting with the classical clarifications of Gödel-Church-Kleene and ending with the newer ones of A.A. Markov,... A.N. Kolmogorov ... . Accordingly, the concept of constructive algebra allows for a number of possible clarifications." A.I. Mal'tsev himself chose ("A.I. Mal'tsev's constructivism in algebra") the approach outlined by A.N. Kolmogorov with one of the initial tasks of describing all possible numberings with the properties of Gödel numbering and relying on recursive functions within the framework of classical logic and set theory ("... it would be interesting to study in some sense all numberings of partially recursive functions..." [29]). In the Siberian mathematical school of academicians Yu. L. Ershov and S. S. Goncharov, the approach received impressive development [30-33]. However, the approach of A. A. Markov (A. A. Markov's constructivism), mentioned by A. I. Maltsev, which does not use the concept of a set, turns out to be in demand in connection with the acute practical problem of artificial intelligence modeling elements of the phenomenon of consciousness on a computer. When designing, manufacturing (and loading programs) computing devices, we are forced to do without the physical implementation of infinity, such a powerful tool of our consciousness in mathematics. In this fundamental problem, it is necessary to follow, among other things, the important methodological principle of science [9] mentioned in the introduction, which is consistent with our systemic approach: do not adjust the formulation of the problem to the usual methods of solution, but carry out a "search for methods corresponding to the original substantive problem". Methodological details of our systemic approach can be found in [34], and here, speaking of algebra, we can limit ourselves to the following. For the sake of certainty, we will rely on the definition of algebra by generators and defining relations.

The presentation we are conducting allows us to distinguish each of the three types of constructivity in algebra: ***general system constructivity***: a finite alphabet is specified, a finite set of rules for constructing words (it is possible to construct individual elements of algebra from generators, use corelations) and statements, a set of tools used by researchers for direct establishment (graphic equality of letters or graphic difference of letters, etc.) of the truth of statements, other truth values, except for "true", are not introduced, the concept of the totality of all elements of algebra, logic is not considered; ***constructivity according to A.A. Markov***: constructive logic, constructed in [8], is added to general system constructivity to prove the truth of a number of statements, the theory of a particular algebra is constructed within the framework of the theory of words in alphabets, developed in [8]; ***constructivity according to A.I. Maltsev***: to the general system constructivity are added the elements of the theory of recursive functions, classical logic and set theory, which includes the concept of infinity, necessary for numbered algebras (which are constructed using generators and corelations). Our presentation is carried out in the usual language accepted in category theory, but it is easy to single out the part related to the general system constructivity, which will allow the work to be used in possible modeling for physical computing devices.

## 4 Categorical Splices and Higher Categories

The formalism of categorical splices is quite general, since theorems similar to the above hold (see [25], where the corresponding categorical splices are constructed) for the simplicial category and a number of higher categories based on globular sets, for categorical objects in the category of small categories Cat (double categories) and a number of higher categories based on cubic sets, as well as for a number of other various higher categories (week multiple categories) studied in [5]. A category is by definition a partial algebra with a binary associative operation of multiplication; in addition to categories, suitable categorical splices are constructed for an arbitrary universal algebra, which was done in [25] immediately for the algebraic theory according to Lawvere [35]. By definition, this is a small category with finite products, each object of which is represented as a power of an object , and the equality holds . By a usual algebraic theory we mean a first-order theory with equality, a signature of function letters (n-ary operations) and several axioms or identities (atomic formulas s=t with terms s,t). The well-known ambiguity of defining an algebraic theory by identities and operations (for example, a group can be defined in addition to the usual three operations of associative multiplication, taking the inverse element and unity, by two operations of double division and unity,

or even by just one operation of division with identities) is overcome in Lawvere's approach. For an algebraic theory T, its syntactic category ST is constructed, which is a category with finite products and an algebraic theory according to Lawvere. Thus, universal algebras in the above sense are models of the corresponding categorical splices. A generalization of Lawvere's theory are the PROPs introduced by MacLane [18], which by definition are symmetric strict monoidal categories whose objects are integers, the tensor product of which on objects coincides with their sum. Lawvere's algebraic theories are props. As indicated in the introduction, props are used for categorical models of a number of networks (electrical, Petri nets, neural networks) [36], [37]. In our approach, as is clear from the above, they represent a special case of models of suitable categorical splices.

## 5    MODELING NEURAL NETWORKS WITH CATEGORICAL SPLICES AND CATEGORICAL SYSTEMS

By neurons and their networks we mean interacting living neurons, as well as their traditional models in the form of artificial neural networks of arbitrary topology. In addition to the above-mentioned modeling of neural networks with props, there are categorical models that use ordinary categories with objects in the form of neurons [38], a similar representation of a neural network is common among neurobiologists (see K.V. Anokhin's questions to the author during the author's report at the Academic Council of the P.K. Anokhin Institute of Normal Physiology of the Russian Academy of Sciences, February 2017, 57th minute of the video https://www.youtube.com/watch?v=s3oTaSt4w0E , apparently, the author proposed to model neurons with polyarrows for the first time). Definition. Let a categorical splice be given with a set of convolutions available for it. A formal categorical system is any comb of this categorical gluing. If a comb has no internal part, then the system is called simple. If there is an internal comb, the system is represented as a corresponding convolution of other combs, called subsystems of this categorical system, which in this case is called a composite system. Let us emphasize that the concept of a simple and composite system reflects the natural procedure of constructing a system from other systems (they become subsystems) using convolution. This differs from the concepts of, for example, a simple group and a group with subgroups. The Mesarovich system corresponds to a simple categorical system with two types of sorts (inputs and outputs). For this systems it is sufficient to consider a special case of categorical gluings called convolutional polycategories. The combs of a convolutional polycategory, as a categorical splice, correspond to sets of polyarrows. In this section, we will define convolutional polycategories and consider the representation of artificial neural networks of arbitrary topology as convolutional polycategories with crown-type convolutions. We will also extend the neuron model that uses convolutional polycategories to a splice-based neuron model, which makes it possible to take into account non-spike communications of neurons among themselves and with other cells.

Definition. A categorical splice is called a convolution polycategory if it has exactly two types of variable sorts ("inputs" and "outputs"), a projection algebra of the form $C_i C_j = C_j, i, j = 1, 2, 3, ...$ with convolutions connecting outputs to inputs. If the generators of a categorical splice are connected combs with one output and several inputs with convolutions having one input and several outputs, then such a categorical splice is called a convolution multicategory. Convolution multicategories generalize ordinary multicategories [14-16], and are also used to model neural networks, as indicated below.

We will briefly touch upon the issue of dualities in categorical splices, which we will need when constructing a precise concept for the "conjugate graph" introduced by S. Osovsky for artificial neural networks. Duality of mutual replacement of sorts. A classic example of this duality is the transition from a category to a dual category, carried out (in an intuitive sense) by the operation of replacing the direction of arrows. If in each comb of this categorical splise we replace two different sorts in the outercomb with each other and replace the corresponding sorts in the convolutions and the inner comb, then we will obtain a new comb, we will call this operation the duality operation by sorts. A new categorical splice consists of such combs, called a dual by sorts categorical splice to the original categorical splice. For the case of convolutional polycategories, the transition to a polycategory dual to sorts, as in ordinary categories, is reduced (in an intuitive sense) to replacing the directions of polyarrows or replacing inputs with outputs and vice versa. Categorical reasoning is very cumbersome, we will consider duality by sorts for the categorical splices already discussed above, modeling ordinary categories. We will write out in graphic form the external comb, consisting

of two connected combs corresponding to the arrows, and the convolution corresponding to the composition

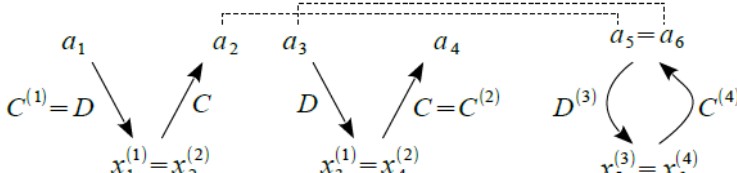

We apply the duality operation, replacing in the full comb obtained from the external combs and the convolution the types 1 and 2, as well as 3 and 4, while leaving the projections unchanged, we have

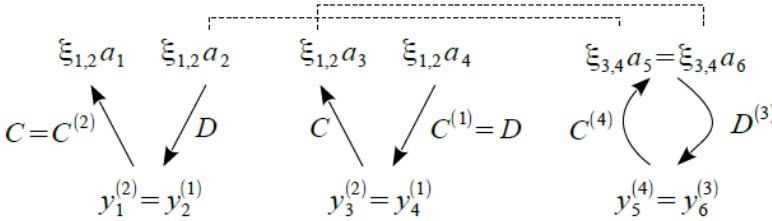

or

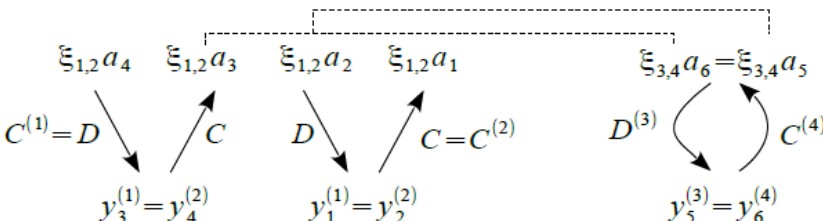

Denoting the convolution predicate s by $K(s, f, g, h)$, we obtain that the duality operation with the projections unchanged led to a change of places of connected combs in their ordered pair and a new formal predicate $K'(s, f, g, h) = K(s, g, f, h)$.

In the general case, the formal predicate $K(s, f, h)$ for two combs s and f connected by equalities of projections, under the duality operation goes over to another predicate $K'(s, f, h) = K(s, gf, h)$, where g is an appropriate permutation of connected external combs of the comb f. ***Duality of mutual replacement of combs and convolutions of a complete comb.*** A new comb can be constructed from a given complete comb of a categorical splice by mutual replacement of the outer and inner convolutions with the outer and inner combs of the complete comb. With such a replacement, a corresponding (as in the operations $M_l, M_r, M_b$) change of sorts occurs. The complete combs transformed in this way constitute a new categorical splice, called a categorical splice dual to the original categorical splice in combs and convolutions.

For a convolutional multicategory, the following theorem is true. Theorem. Let a convolutional multicategory and its full ridge be given. Then performing the duality operation on sorts and the duality operation on combs and convolutions transforms the specified full ridge into another full ridge of a new convolutional multicategory, in which each connected outer ridge transforms into a connected outer convolution and vice versa.

Ordinary multicategories are studied for the case of associativity of the composition of multiarrows, a convolutional polycategory may not satisfy the analog of the associativity condition, it should be introduced additionally if necessary, as well as the condition on convolutions, for their compliance with the usual composition. Models of formal theory with such conditions (the formulation of which

is very cumbersome) on convolutions are called associative compositional convolutional multicategories. As is known [15], for multicategories, changing the directions of arrows does not lead to duality similar to the duality of categories. The given theorem gives a natural version of duality for multicategories. The construction of dual categorical splices plays no less an important role in their theory than in ordinary category theory. One of the key theorems of category theory, justifying the duality principle [39], has the form (see [28]) Theorem. Let A be any well-formed formula provable in formal category theory, then the dual formula for A is also provable. If we embed categorical splices in a suitable first-order theory with classical logic, then a similar theorem will be true as well. Theorem. Let A be any well-formed formula provable in the formal theory of categorical splices, then the dual by sorts and dual by ridges and convolutions formulas for A are also provable.

Let us move on to defining a new splice model of neurons and their networks. The polycategorical model neuron was introduced by the author. The inputs and outputs of polyarrows representing neurons model the spike propagation paths. However, intercellular communications are much richer than spike activity. Many studies have found that other types of connections between neurons influence spiking activity. Models of neurons have emerged [3] that take into account such interactions. Definition. Let a categorical splice be given, having two sorts of variables, which we will call inputs (in) and outputs (out), and n sorts of variables, which we will call sorts of non-spike communication channels. Then a neuron is called each connected comb of the categorical splice, and the categorical splice itself with the existing convolutions is called a categorical splice neural network.

Further we will talk about formal categorical neural networks within the framework of first-order formal theories with classical logic and with interpretation in sets. Since categorical splices include, as special cases, convolutional polycategories and higher categories, this definition, taking into account the presence of corresponding convolutions in the combs, is of a very general nature and has a rich toolkit for modeling highly complex connections between neurons interacting not only by spikes. Next, we consider the modeling of traditional artificial neural networks with convolutional multicategories. One of the main existing generally accepted approaches in the theory of neural networks, namely, PDP (Parallel Distributed Processing), was developed in the 1980s by a group of scientists - physiologists, psychologists, mathematicians, computer scientists, whose members included Nobel laureate Francis Crick. The PDP neural network model contains three types of elements, a neuron or soma (the processing element, now it is a processor or transputer), an axon (the transmission line of signals coming from the soma) and a synapse (a "junction" that converts the signal from the axon into a suitable signal for the soma). The neural PDP network in its simplest version forms a "graph" with nodes in the form of neurons with the following properties: (a) only one line approaches each synapse; (b) lines from different neurons approach different synapses on a given neuron; (c) transmission lines do not branch and are ordinary arcs of the graph. Property (c), taken from [39] p.274, actually requires that a neuron have multiple outputs, this is wrong not only because further in the text [39] a neuron produces only one signal, but also because it obscures the most important property of living neural networks, when one output is connected to several inputs of other neurons. Here, a variant of convolution is clearly used, which differs from the method of connecting arrows using compositions known in category theory. In the graphical notations from [40], where the PDP model is described, point (c) is intended to replace

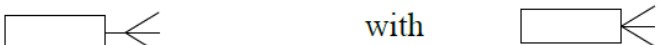

which fails; in the example from [39] p.275 in the figure below, branching of lines appears again.

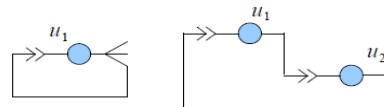

In our approach, we associate a neuron in this PDP model with a polyarrow with inputs in the form of synapses and an output in the form of a line, which, like an axon, splits into several lines going

to other neurons. The concept of a graph includes, by definition, two sets, a set of vertices and a set of pairs of vertices. Usually, neural network engineers continue to talk about a neural network as a graph, but when they are presented with a branch point from a neural network along with the definition of a graph (with which they agree!) that is not included in the graph, they get into a difficult situation: a neural network is not a graph in the usual sense, as this corresponds to the given definition. And the attempt (in the figure) to "remove" this branch point into a neuron is not accidental. The main modern standard of neuroinformatics (see [41]) is formed by neurons that calculate the value

$$a = f(Wx+b), x = (x_1, ..., x_k) \in \{0,1\}^k, a \in \{0,1\}$$
$$W = (w_1, ..., w_k)^T, w_i, b \in \mathbb{R}, f : \mathbb{R} \to \{0,1\} \ .$$

, .

Neurons in a neural network are connected to each other in such a way that from a single neuron output (axon), the connection branches out to several inputs of other neurons. Thus, a neuron n is represented as a function of many variables $n : 0, 1^k \to 0, 1$ . For a neural network, the connections of neurons can be modeled by a convolution in an associative compositional convolutional multicategory according to the following theorem.

Theorem. (modeling neural networks) Let there be an artificial neural network with neurons $n$ : $0, 1^k \to 0, 1$ having several inputs ($k$=1,2,3, ... ) and one output, with its own activation function for each neuron, with signals coming to the input of the neuron's synapses from a set b. The connections of neurons are carried out by the existing output, which branches into a finite number $m$=1,2,3, ... of lines connecting with the inputs of other neurons. Then the neural networks built from the specified neurons form an associative compositional convolutional multicategory.

The specified categorical representation of a neural network helps to eliminate a number of inaccuracies in their theory. For example , relying on the properties and concept of duality of polycategories, we present correct schemes of the intuitive method of S. Osovsky [2] popular among engineers for calculating partial derivatives used in the method of backpropagation of error. Let us consider a general unidirectional multilayer artificial neural network in the notations from [2] (p. 55).

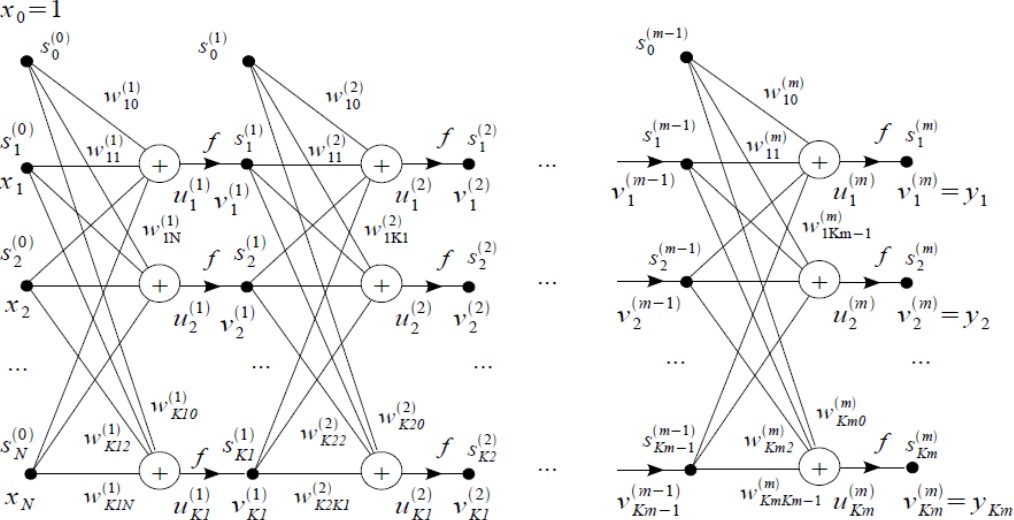

According to the theorem on the representation of a neural network by a convolutional multicategory, there is a categorical model of the specified neural network, by applying duality operations to which one can construct the following dual full comb of the dual convolutional multicategory.

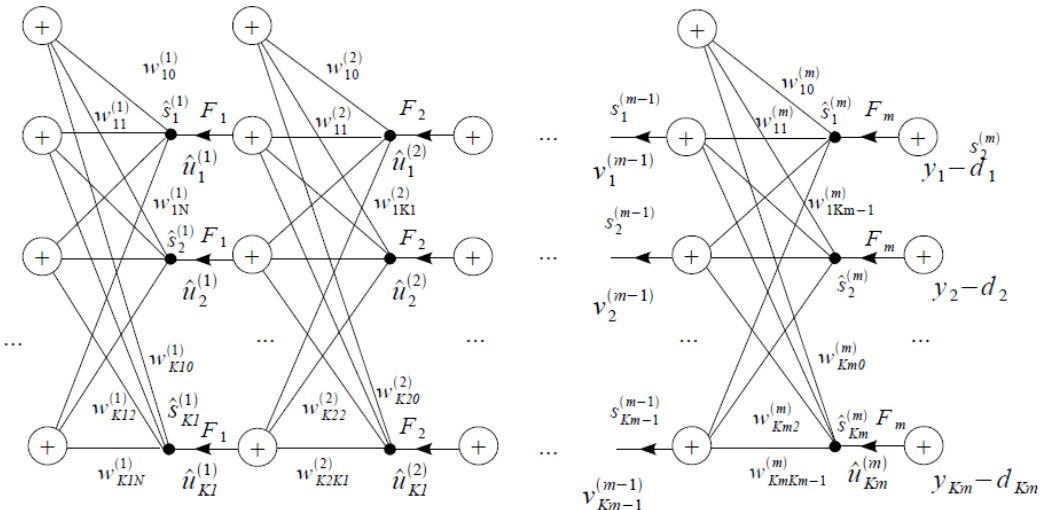

We obtain the exact object that S. Osovsky built intuitively. He called it a "conjugate graph". Note that our scheme introduces additional neurons compared to Osovsky's scheme ([2] Fig. 3.7., p. 55), into which the convolutions are translated according to duality transformations.

## 6 CONCLUSION

In the categorical theory of systems developed by the author, three levels of constructivism naturally arise that are important for the development of artificial intelligence: general system constructivism, constructivism according to A.A. Markov, and constructivism according to A.I. Maltsev. According to the system paradigm of the categorical theory of systems, neural networks, both artificial neural networks and networks of living neurons and other cells, are systems modeled by categorical splices and convolutional polycategories. The presented models are not only adequate to the systemic nature of traditional neural networks of arbitrary topology, but also offer a formalism for higher-order networks that correspond to analogs of higher categories, and for taking into account the non-spike activity of neurons both among themselves and with other types of cells. In addition to the duality known in category theory (construction of a dual category), categorical splices has another type of duality, which is considered in the paper. The combination of these types of duality provides a natural solution to the problem of constructing duality for multicategories. The proposed transition from a convolutional multicategory to its dual multicategory formalizes (as described in the paper) the intuitive construction of a "conjugate graph" for the well-known formulas of S. Osovsky used in calculating the gradient in the backpropagation method.

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
