# OpenReview forum: "Categorical model of neural networks"
_mathai.club/MathAI/2025/Conference — MathAI 2025 Oral_

### Official Review · Reviewer_BHYz · 2025-02-24
**Review on: Categorical Model of Neural Networks**

**Rating:** 7
**Confidence:** 4

**Review:**

Strong points:
1) The paper fits the conference "MathAI" agenda perfectly.
2) The paper provides sound mathematical foundation for the technology of neural networks from perspective of categorical algebra

Weak points:
1) Paper is not formatted according to the conference template which MUST be fixed.
2) References are not well formatted, example: "... ones of A.A. Markov,... A.N. Kolmogorov ... " - MUST be fixed.
3) No conclusion is made - MUST be fixed.
4) No practical implications and possible applications are pointed out.
5) No experimental support is provided to validate the mathematical theory.

P.S. Reference to https://arxiv.org/pdf/1901.01341 might be relevant and useful.

---

### Official Review · Reviewer_EK3x · 2025-02-26
**Review on: Categorical Model of Neural Networks**

**Rating:** 8
**Confidence:** 4

**Review:**

Pros
The paper fits the conference "MathAI" agenda.
Originality and significance: An original categorical theory of systems is developed that allows naturally model not only traditional artificial neural networks of arbitrary topology, but also networks of living neurons. In particular, neural networks models as categorical systems was developed. The paper presents a number of new definitions and results, provides proofs of theorems, and also a brief overview of works on categorical topics in systems theory and neural networks.
Three types of constructivity in algebra is considered: general system constructivity, constructivity according to A.A. Markov and constructivity according to A.I. Maltsev.
In addition to the modeling of neural networks, there are categorical models that use ordinary categories with objects in the form of neurons, a similar representation of a neural network is common among neurobiologists.

Cons:
Paper and references are not formatted according with the conference template.
Conclusion MUST be added.
Definitions and theorems must be formatting such that the end of them be clear.

---

### Official Review · Reviewer_ArK5 · 2025-02-27
**Categorical model of neural networks**

**Rating:** 8
**Confidence:** 4

**Review:**

The paper introduces an original categorical theory of systems capable of modeling both artificial and biological neural networks. New definitions, results, and theorem proofs are presented, along with a review of prior research. Various types of algebraic constructivity are explored. The models employ standard categories with neuron-like objects, making them relevant to neurobiology. However, the paper would benefit from the inclusion of practical applications and experimental verification of the theory.

---

### Official Review · Reviewer_Rt6C · 2025-02-27
**Categorical model of neural networks**

**Rating:** 7
**Confidence:** 4

**Review:**

The article introduces a neural network model based on category theory by incorporating the concepts of convolutional polycategories and categorical splices, aiming to identify the structure and function of neural networks from a systemic perspective. The proposed model demonstrates strong innovation and provides a rich theoretical foundation along with illustrative explanations.
However, while the article offers a theoretical framework, the explanation of the example regarding the conjugate graph is somewhat brief and may require deeper elaboration for better understanding. Modifying the article's format by adding formulas, reducing text, and including a conclusion would enhance its quality.

---

### Decision · Program_Chairs · 2025-03-08

**Decision:**

Accept (Oral)

**Comment:**

Your article has been accepted and you can make a presentation on the article. All articles will be sorted by rating and within the available conference places one author from each article will be invited. If there are not enough places, then you will either have the opportunity to present remotely or come at your own expense!